# Management of Spontaneous Crystalline Lens Luxation in a Patient Diagnosed with Takayasu’s Disease

**DOI:** 10.3390/diagnostics13081400

**Published:** 2023-04-12

**Authors:** Elvia Mastrogiuseppe, Maria Pia Pirraglia, Lorenzo Sampalmieri, Ludovico Iannetti, Alessandro Beccia, Magda Gharbiya

**Affiliations:** Ophthalmology Unit, Head and Neck Department, Policlinico Umberto I University Hospital, Department of Sense Organs, Sapienza University of Rome, 00185 Roma, Italy

**Keywords:** spontaneous lens luxation, Takayasu arteritis, intrascleral fixation of intraocular lens, Yamane’s double-needle technique

## Abstract

Takayasu’s disease is a chronic granulomatous arteriopathy that affects large vessels and their major branches. Nonspecific symptoms characterize the early phase, whereas findings of arterial occlusion and aneurysmal formation become manifest later. Ocular signs typically refer to retinal vascular involvement, as Takayasu arteritis or hypertensive retinopathy. We report a case of a 63-year-old woman suffering from Takayasu arteritis that complained of sudden onset of blurred vision in her left eye due to crystalline lens luxation in the vitreous cavity. The patient’s past medical history was unremarkable for trauma, personal or familiar collagenopathies. Prompt surgical management was performed and the patient reached 0 LogMAR seven days after surgery. Our case illustrates the concomitant occurrence, never reported before, of two rare conditions in the same patient, namely, Takayasu arteritis and spontaneous lens dislocation. Further research and future knowledge are needed to explain whether Takayasu arteritis could obliquely injure zonular or fibrillar structures and whether these features may be possibly related.

## 1. Introduction

Takayasu arteritis (TA) is a chronic inflammatory arteriopathy of unknown etiology and variable course, leading to multiorgan damage due to tissue hypoperfusion. Described for the first time in 1908 by Takayasu, an ophthalmologist [1], TA is also named pulseless disease, due to the involvement of the aorta and its proximal branches.

TA has a worldwide prevalence, although it mostly occurs in women during the second to third decade of life, most commonly in Asia and Africa [2]. The systemic nature of TA leads to a varying degree of arterial injuries, from thickening to aneurysm and thrombus formation. During the pre-vasculitis phase, nonspecific symptoms, such as malaise, fever, headache, arthralgias, and weight loss, make diagnosis challenging. In the late stages, symptoms related to vascular involvement become clear and patients present with a wide variety of clinical syndromes such as pulseless extremities, limb claudication, blood pressure discrepancy, vascular bruits, hypertension, Raynaud’s syndrome, aortic regurgitation, stroke and coronary heart disease, dermatologic and neurological manifestations [3]. We report a case of a 63-year-old woman affected by TA presenting atraumatic crystalline lens luxation in her left eye (LE). Written informed consent was obtained from the patient.

## 2. Case Presentation

A 63-year-old woman presented at our clinic complaining of blurred vision in her LE for two days, associated with transient ocular puncture pain. Her past ophthalmological history was unremarkable for facial or orbital trauma, previous surgical procedures or intravitreal injections.

One year before the patient reported an episode of claudication in her left arm. Computed tomography (CT) angiography revealed circumferential wall thickening in the left axillary artery associated with stenosis and distal occlusion consistent with inflammatory diseases. Positron emission tomography (PET)/CT images showed increased uptake of 18F-fluorodeoxyglucose (FDG) along the ascending aorta, aortic arch, descending tract of thoracic aorta and main branches of brachiocephalic artery, common carotid, axillary and subclavian arteries. Erythrocyte sedimentation rate (ESR) was 79.0 mm/h (cut-off 25 mm/h) and only antinuclear antibodies (ANA) were detected with an antibody titer equal to 1:80 and a fine speckled pattern. Based on the instrumental, laboratory and clinical results, the patient was diagnosed with TA. She was treated with six 1gr cyclophosphamide intravenous infusions administered monthly. The dosage was then tapered to one 50 mg tablet twice a day, 5/7 days a week for 4 months. She was also given prednisone 25 mg, one tablet per day, which was tapered down to two 5 mg tablets per day. Six months before she had reported lower back pain and high blood pressure due to left renal artery involvement, managed with prednisone 75 mg per day then tapered down to 18.75 mg per day, furosemide 25 mg per day, olmesartan medoxomil/amlodipine 25/5 mg per day. Her past medical history was positive for polymyalgia rheumatica, allergic asthma, monoclonal gammopathy, thalassemia and iatrogenic hypothyroidism. The patient’s home therapy was prednisone 25 mg per day, furosemide 25 mg per day, olmesartan medoxomil/amlodipine 25/5 mg per day, clopidogrel 75 mg per day, atorvastatin 10 mg per day, levothyroxine 75 mg per day, pantoprazole 40 mg per day, alendronate 70 mg per day and cholecalciferol 10,000 UI 14 gtt per day.

Seven days before the patient underwent an ophthalmological evaluation which revealed 0 LogMAR best corrected visual acuity (BCVA) in the right eye (RE) and 0.22 LogMAR in the LE. On slit lamp evaluation, the anterior segment appeared remarkable only for mild nuclear, cortical and subcapsular cataracts in both eyes (BEs), slightly more in the LE than in the RE. Intraocular pressure (IOP) was 18 mmHg in the RE and 22 mmHg in the LE. On indirect ophthalmoscopy, arterial thinning and moderate vascular tortuosity were noted in BEs. The next eye examination was scheduled after a visual field test and retinal nerve fiber layer (RNFL) evaluation using optical coherence tomography (OCT).

During our ocular examination, her BCVA was 0 LogMAR in the RE and finger counting in the LE. The anterior segment appeared unchanged in the RE (Figure 1a), however aphakia and vitreous prolapse in the anterior chamber were detected in the LE (Figure 1b) without any sign of trauma or inflammation. The Anterior Segment Optical Coherence Tomography (AS-OCT) confirmed these clinical findings (Figure 2). IOP was 14 mmHg in BEs. Fundus examination confirmed vascular thinning and tortuosity in BEs with crystalline lens luxation in the vitreous chamber in the LE (Figure 3 and Figure 4). Laboratory results revealed normal erythrocyte sedimentation rate (ESR) amounted to 2.0 mm/h (cut-off 35 mm/h).

The patient underwent prompt vitreoretinal surgery. Removal of the lens was performed using a vitrectomy cutter handpiece with proportional vacuum and cutting setting after 25-gauge pars plana vitrectomy. An intrascleral three-piece intraocular lens (IOL) KOWA PU6AS (power +24.50 D) was implanted using flanged fixation with 30-gauge double-needle using the Yamane technique (Figure 5) [4]. The day after surgery, the postoperative course was regular with clear cornea, slight reaction in the anterior chamber (flare 1+ and cells 1+), IOL well positioned and the patient was given therapy with steroid and antibiotic drops (dexamethasone four times daily and chloramphenicol four times daily). One week later, BCVA reached 0 LogMAR, the cornea was transparent and aqueous humor clear, IOL was centered and well-placed and IOP was 17 mmHg in the LE (Figure 6). Fundus ophthalmoscopy was unremarkable. The patient was advised to discontinue antibiotic drops and to taper down steroid drops over four weeks time. One month after surgery, BCVA remained at 0 LogMAR, the ocular conditions were stable, and IOL was centered and well-placed.

## 3. Discussion

TA is a rare systemic inflammatory disease affecting large arteries with unclear pathogenesis. Although diagnostic criteria for TA still need to be validated, the presence of hallmark signs and symptoms is suggestive of TA, such as those included in Ishikawa’s criteria [5]: pulseless, asynchronous arterial pulses or blood pressure in the arms, unobtainable blood pressure, easy limb fatigability or pain, and minor signs as unexplained fever or high erythrocyte sedimentation rate (ESR) or neck pain associated with transient amaurosis or blurred vision or syncope, dyspnea or palpitations or both, hypertension or aortic regurgitation [6]. Nowadays, the diagnosis of TA is based on the sum of clinical presentation and imaging results; histopathologic confirmation can be obtained only during vascular surgery [3].

Ocular involvement occurs in up to 68% of patients and retinal manifestations are the most frequently reported. Typical retinal patterns in TA include: Takayasu’s retinopathy (TR); hypotensive retinopathy following hypoperfusion of the eye due to carotid involvement and hypertensive retinopathy, which is secondary to renal artery stenosis and characterized by the dilatation of small vessels, capillary microaneurysms, arteriovenous anastomosis and other ocular complications [7]. TR has been classified into four stages: stage one is characterized by distension of veins; stage two is characterized by capillary microaneurysm formation; stage three is characterized by arterio-venous anastomosis; stage four is characterized by the presence of ocular complications such as cataract, rubeosis iridis, retinal ischemia, neovascularization and vitreous hemorrhage. Other ischemic manifestations have been described infrequently, such as anterior ischemic optic neuropathy, central retinal artery occlusion and ocular ischemic syndrome [8]. Additionally, these ischemic manifestations have been reported as bilateral by Pallangyo et al. [3]. Rarely, recurrent nodular anterior scleritis has been reported in association with TA as the first, earliest and isolated ocular and systemic sign [9].

In ectopia lentis, also known as lens subluxation or dislocation, the lens is untethered: if the dislocated lens remains within the patellar fossa it is considered subluxated, otherwise an entirely dislocated lens is described as luxated. Usually, ectopia lentis is associated with the rupture of the zonular fibers. Most cases of lens luxation are due to a direct ocular trauma [10]. However, fragmentation of zonular fibers, and the subsequent destabilization of the lens, are also a feature of some common ocular diseases. The most well-known clinical condition is the exfoliation syndrome (XFS). XFS is characterized by the deposition of insoluble fibrillar aggregates on the surface of tissues throughout the anterior segment, including the corneal endothelium, iris, lens and zonule [11]. In our case, none of the hallmarks of XFS were found in either the aphakic eye on biomicroscopy and gonioscopy evaluation or in the fellow eye.

Ectopia lentis is associated with several inherited conditions, including familiar collagenopathies such as Marfan syndrome, Weill–Marchesani syndrome (WMS), and homocystinuria. Marfan syndrome (MS) is an autosomal dominant (AD) disease characterized by cardiac, musculoskeletal and ophthalmological manifestations. Classically, patients have arachnodactyly and a tall slender appearance. Ectopia lentis is commonly subluxated superotemporally [12]. WMS is a rare connective tissue disorder characterized by brachydactyly, short stature, joint stiffness, cardiovascular abnormalities and eye anomalies including microspherophakia, cataracts, ectopia lentis, myopia and secondary glaucoma [13]. Homocystinuria is a rare autosomal recessive metabolic disease caused by cystathionine-b-synthase deficiency characterized by high levels of plasmatic homocysteine concentration, ocular disorders, skeletal abnormalities, developmental delays and thromboembolic events [12]. In this case, the lens is typically dislocated inferonasally and it occurs in almost 90% of patients. Bilateral ectopia occurs in 60% of cases and in nearly 100% of cases by the age of 25 [14]. Our patient had none of these features and her familiar history was unremarkable for hereditary disease.

MS and WMS are due to mutations in the FBN1 gene, which encodes for fibrillin, a protein that provides structural support and elasticity to ocular connective tissues [15]. However, the biochemical mechanism underlying homocystinuria is not yet fully explained. Some hypotheses include: defects in fibril disulfide bridges as a basis for lens dislocation, or the deficiency in cystathionine-b-synthase could affect the nutritional metabolism of the lens zonule, causing its degeneration and rupture. Furthermore, elevated homocysteine levels may interfere with the cross-linking of sulfhydryl groups in elastin [12]. 

Ehlers–Danlos syndrome (EDS) is a rare AD heritable condition affecting connective tissue, characterized by skin hyper-extensibility, tissue fragility and generalized joint hypermobility. COL5A1 or COL5A2 genes, which normally code for collagen chains α1(V) and α2 (V), are mutated and an abnormal coding results in structurally and functionally defective type V collagen [16]. Hashimoto et al. [17] reported a case of TA associated with EDS, causing multiple aneurysms and arterial dissection at a young age owing to a mutation in the gene for type III collagen, COL3A1, without any ocular signs or symptoms. 

Lens dislocation combined with red eye history has also been related to the latent or tertiary stage of syphilis, the most common stages of ocular involvement during syphilitic infection [14,18]. Anubha et al. [19] reported a case of bilateral infective nodular scleritis, anterior uveitis and aphakia with unilateral perforated peripheral ulcerative keratitis in a man with genital ulcers. No signs of ulcers were detected in our patient, venereal disease research laboratory (VDRL) test and fluorescent treponemal antibody absorption (FTA-ABS) test were negative and syphilis was ruled out.

Subluxation and luxation of crystalline lens in vitreous, as well as all conditions of aphakia, need to be resolved for visual purposes using a scleral-fixated intraocular lens. Currently, surgical approaches for aphakia correction include anterior chamber IOL, iris claw IOL, sutured and sutureless techniques of scleral-fixated intraocular lens [4]. Guidelines for IOL implantation in the absence of capsular support remain controversial. We decided to perform the Yamane technique, which is a new method of sutureless scleral-fixated IOL that is relatively safe and easy. Furthermore, flanged intrascleral IOL fixation provides more stability of the haptics, preventing their dislocation compared with conventional transscleral suturing [4].

In our case, we detected the association of two rare conditions, namely, TA and spontaneous lens luxation. Based on the uncertain pathogenesis of TA, one possible speculative hypothesis is that chronic granulomatous inflammation, affecting branches of the carotid artery, may cause an impaired nutritional intake downstream in the zonule vascularization, resulting in permanent fibrillar vulnerability and weakness. We hypothesize that this damage may occur where blood flow is more susceptible, due to an hypoperfusion along major vascular branches that might lead to ischemia in the tissues vascularized by small vessels in favor of more precious tissues, such as the retina. Anterior segment involvement could be avoided thanks to the role of the major arterial circle of the iris and its anastomosis with anterior ciliary arteries, in which case a more extended hypoperfusion could be required to cause long-term ischemic injuries. 

Zeng Y et al. [20] reported that acute and chronic visual loss could be secondary to TR at stage two with no pathologic findings on fluor angiography imaging. They also reported that visual improvement could be obtained after surgical vascular approaches. Gain in retinal function could be achieved after vascular reperfusion, thereby providing proof of focal undetectable microvascular alterations.

Small vessels could also be the direct target of the disease, as assumed by Das et al. [21]. Possible autoimmune processes, including immune-regulatory genes and inflammatory cytokines involved in TA, could also play a role in the initial pathogenic stimulus targeted to molecules or subcellular elements of zonular components. In this regard, patients with TA have been found to have increased serum levels of IL-6, which correlate with disease activity, increased levels of IL-6 expression in aortic tissue, suggesting an intravascular synthesis. IL-6 is known to play multiple roles, such as activating B cells, enhancing T cell cytotoxicity, promoting NK cell activity, activating matrix metalloproteinases (MMPs), fibroblast proliferation and acute-phase protein synthesis [22,23,24]. Alibaz-Oner et al. also found higher serum levels of IFN-γ and an increase in other cytokines, such as IL-8, IL-9, IL-17, IL-18 and tumor necrosis factor (TNF), in patients suffering from TA compared with healthy controls [25].

## 4. Conclusions

TA is a rare idiopathic systemic inflammatory disease with life-threatening potential. To our knowledge, this is the first reported case of spontaneous crystalline lens luxation associated with TA. Rapid and adequate management associated with a correct and complete diagnostic path is necessary to reach anatomical and functional success. Future challenges include gaining a better understanding of pathogenic patterns to identify novel biomarkers, the effects on macro and microcirculation and, eventually, explaining ocular and systemic involvement. Furthermore, a better disease comprehension could lead to a prompt and adequate therapy, with delayed disease progression, reduction in fatal complications and prevention of blindness.

## Figures and Tables

**Figure 1 diagnostics-13-01400-f001:**
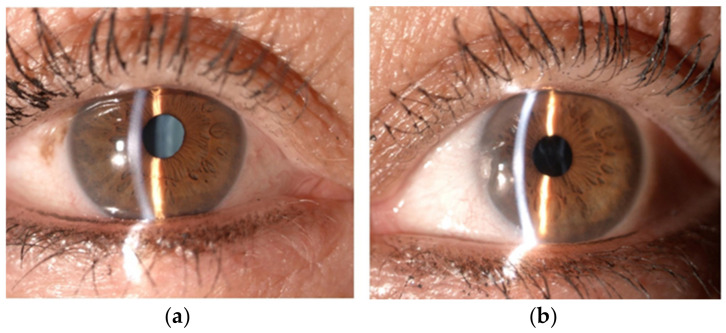
Anterior segment of right eye (**a**) and left eye (**b**).

**Figure 2 diagnostics-13-01400-f002:**
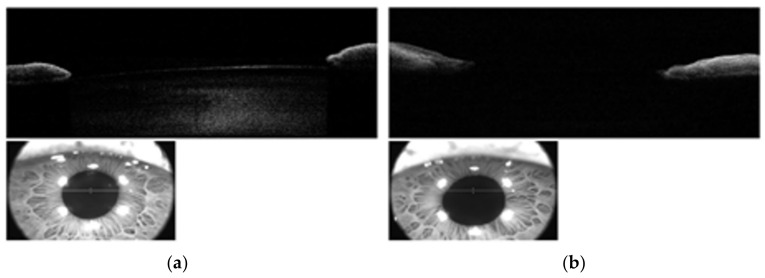
Anterior Segment Optical Coherence Tomography showing anterior capsule and cortex of lens behind the iris in right eye (**a**); same structures were not detectable in left eye (**b**).

**Figure 3 diagnostics-13-01400-f003:**
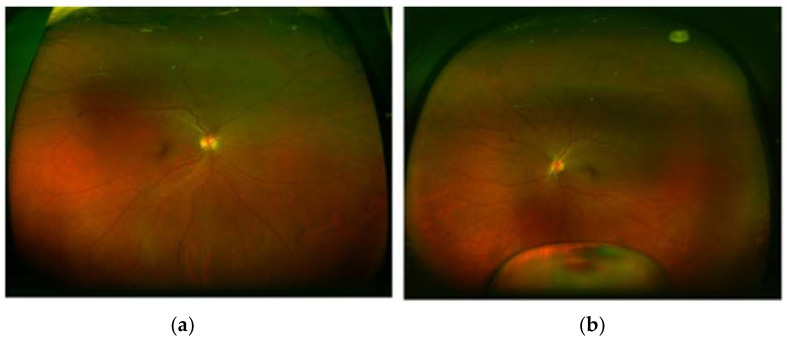
Optos pseudocolor ultra-widefield fundus photograph of right eye (**a**) and left eye (**b**) showing lens dislocation in vitreous cavity.

**Figure 4 diagnostics-13-01400-f004:**
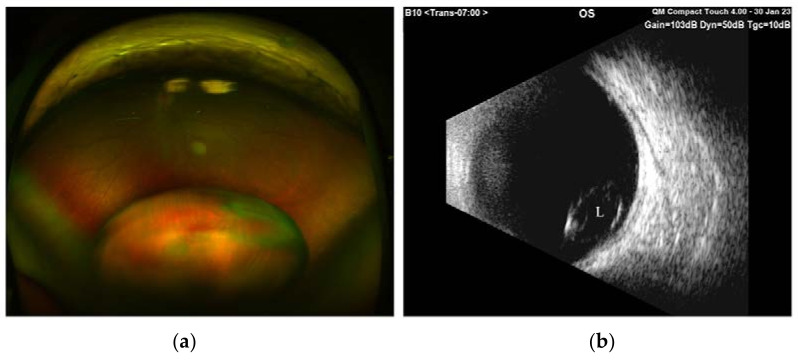
(**a**) Inferior lens luxation in vitreous chamber on optos pseudocolor ultra-widefield fundus photograph; (**b**) ultrasound B-scan showing lens dislocation (L) detected as a biconvex structure with faint internal echoes.

**Figure 5 diagnostics-13-01400-f005:**
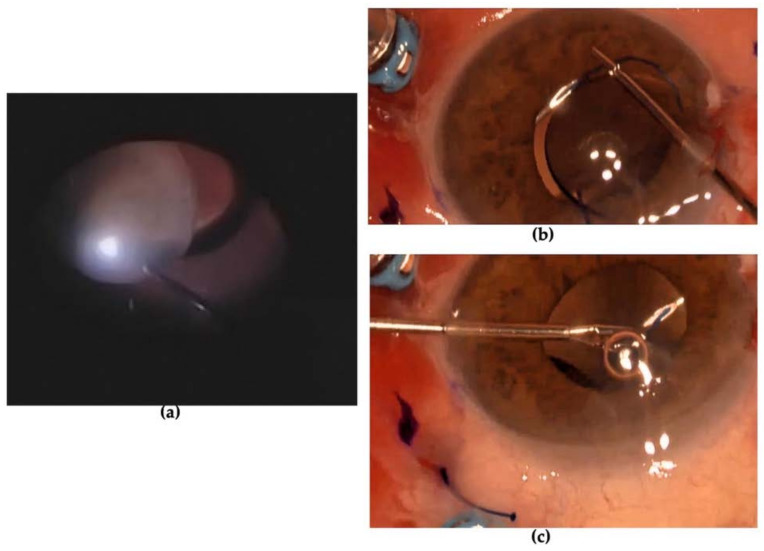
(**a**) Lens removal using vitreous cutters; (**b**) three-piece intraocular lens insertion in the anterior chamber through sclerocorneal tunnel; first haptic was docked into 30-gauge needles inserted under scleral groove; (**c**) first haptic was withdrawn, melted with a cautery and nudged into the scleral tunnels; second haptic was docked into needle.

**Figure 6 diagnostics-13-01400-f006:**
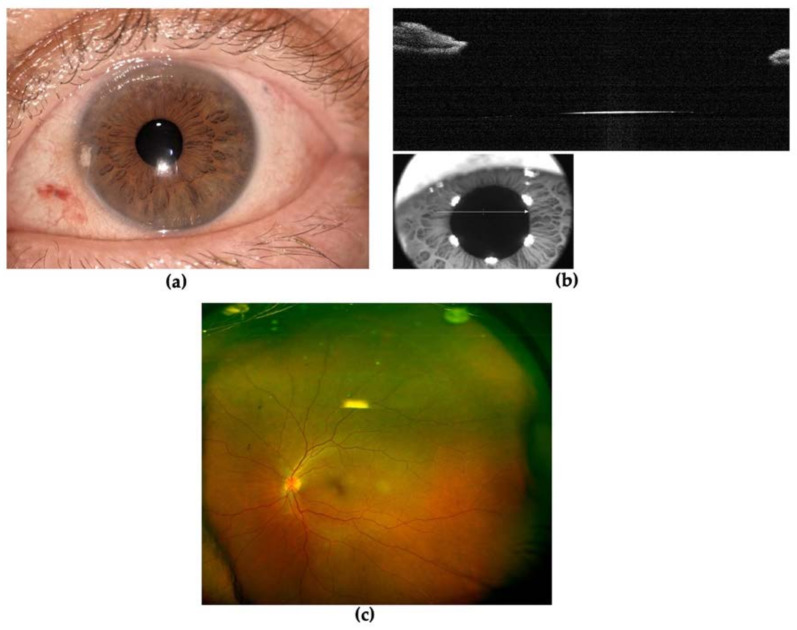
One week after surgery (**a**) anterior segment photography; (**b**) Anterior Segment Optical Coherence Tomography showing intraocular lens optic; (**c**) optos pseudocolor ultra-widefield fundus photograph of left eye.

## Data Availability

The data that support the findings of this study are not publicly available to protect the privacy of the research participant but are available from the corresponding author, E.M.

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
