# Peer review of "Management of Spontaneous Crystalline Lens Luxation in a Patient Diagnosed with Takayasu’s Disease"

_diagnostics, 2023, doi:10.3390/diagnostics13081400_

Round 1

Reviewer 1 Report

This case report is meticulously written and English is satisfactory. 

However, authors should spend more space in explaining the pathogenesis behind their proposed theory. More specifically regarding their comment ""Based on the uncertain pathogenesis of TA, a speculative 195 hypothesis could be that chronic granulomatous inflammation, affecting branches of the 196 carotid artery, may cause an impaired nutritional intake downstream in the zonule vas-197 cularization determining permanent fibrillar vulnerability and weakness"" authors are advised to explain why the inflammation did not cause any haemodynamic alterations to the anterior/posterior segment. Additionally, if there have been any changes to the AS/PS after the surgical correction in terms of long term follow-up. 

Author Response

Dear Reviewer,

Thank you for reviewing our manuscript and for your useful comments. We have explained more extensively the pathogenesis proposed. In particular we may hypothesize that the damage in the zonule could occur where blood flow is more susceptible, due to an hypoperfusion along major vascular branches that might lead ischaemia in the tissues vascularized by small vessels in favour of more precious tissues, such as retina. Anterior segment involvement could be avoided thanks to the role of the major arterial circle of the iris and its anastomosis with anterior ciliar arteries, in which, therefore, a more extended hypoperfusion could be required to cause long term ischemic injuries. We added this part in the text and we added two new references as well.

The follow-up after our surgery of the present case is up to 1 month. Until that moment, no other change to the AS/PS was observed.

Reviewer 2 Report

This is an interesting article reporting the concomitant occurrence of two rare conditions (Takayasu arteritis and spontaneous lens dislocation) in the same patient; no structural revisions are suggested before possible publication.

Line 147: “XPS” should be changed to “XFS”

Line 204: “IL 6” should be changed to “IL-6”

Author Response

Dear Reviewer,

Thank you for reviewing our manuscript and for your comment.

Line 147: “XPS” should be changed to “XFS”

Amended

Line 204: “IL 6” should be changed to “IL-6”

Amended